# PHYTOCHROME C regulation of photoperiodic flowering via *PHOTOPERIOD1* is mediated by *EARLY FLOWERING 3* in *Brachypodium distachyon*

**Daniel P. Woods**[1,2☯]*, **Weiya Li**[3☯], **Richard Sibout**[4,5], **Mingqin Shao**[6], **Debbie Laudencia-Chingcuanco**[7], **John P. Vogel**[6], **Jorge Dubcovsky**[1,2], **Richard M. Amasino**[3,8]*

**1** Dept. Plant Sciences, University of California, Davis, California, United States of America, **2** Howard Hughes Medical Institute, Chevy Chase, Maryland, United States of America, **3** Department of Biochemistry, University of Wisconsin, Madison, Wisconsin, United States of America, **4** Institut Jean-Pierre Bourgin, UMR1318 INRAE-AgroParisTech, Versailles Cedex, France, **5** UR1268 BIA, INRAE, Nantes, France, **6** DOE Joint Genome Institute, Berkeley, California, United States of America, **7** USDA-Agricultural Research Service, Western Regional Research Center, Albany, California, United States of America, **8** United States Department of Energy Great Lakes Bioenergy Research Center, University of Wisconsin, Madison, Wisconsin, United States of America

☯ These authors contributed equally to this work.
* dpwoods@ucdavis.edu (DPW); amasino@biochem.wisc.edu (RMA)

**Data Availability Statement:** All mutant lines are available upon request to the following email

## Abstract

Daylength sensing in many plants is critical for coordinating the timing of flowering with the appropriate season. Temperate climate-adapted grasses such as *Brachypodium distachyon* flower during the spring when days are becoming longer. The photoreceptor PHYTOCHROME C is essential for long-day (LD) flowering in *B. distachyon*. *PHYC* is required for the LD activation of a suite of genes in the photoperiod pathway including *PHOTOPERIOD1* (*PPD1*) that, in turn, result in the activation of *FLOWERING LOCUS T* (*FT1*)/*FLORIGEN*, which causes flowering. Thus, *B. distachyon phyC* mutants are extremely delayed in flowering. Here we show that PHYC-mediated activation of PPD1 occurs via *EARLY FLOWERING 3* (*ELF3*), a component of the evening complex in the circadian clock. The extreme delay of flowering of the *phyC* mutant disappears when combined with an *elf3* loss-of-function mutation. Moreover, the dampened *PPD1* expression in *phyC* mutant plants is elevated in *phyC/elf3* mutant plants consistent with the rapid flowering of the double mutant. We show that loss of *PPD1* function also results in reduced *FT1* expression and extremely delayed flowering consistent with results from wheat and barley. Additionally, *elf3* mutant plants have elevated expression levels of *PPD1*, and we show that overexpression of *ELF3* results in delayed flowering associated with a reduction of *PPD1* and *FT1* expression, indicating that ELF3 represses *PPD1* transcription consistent with previous studies showing that ELF3 binds to the *PPD1* promoter. Indeed, *PPD1* is the main target of ELF3-mediated flowering as *elf3/ppd1* double mutant plants are delayed flowering. Our results indicate that *ELF3* operates downstream from *PHYC* and acts as a repressor of *PPD1* in the photoperiod flowering pathway of *B. distachyon*.

address: front_desk@biochem.wisc.edu
(Department of Biochemistry, University of
Wisconsin) with no restrictions.

**Funding:** This work was funded in part by the US
National Science Foundation (Award IOS-1755224
to RMA) and by the Great Lakes Bioenergy
Research Center, US Department of Energy, Office
of Science, Office of Biological and Environmental
Research (Award No. DE-SC0018409). The work
from JPV, RS, and MS (proposal: 10.46936/
10.25585/60001041) conducted by the U.S.
Department of Energy Joint Genome Institute
(https://ror.org/04xm1d337), a DOE Office of
Science User Facility, is supported by the Office of
Science of the U.S. Department of Energy operated
under Contract No. DE-AC02-05CH11231. DPW
was a Howard Hughes Medical Institute (HHMI)
Fellow of the Life Sciences Research Foundation
which supported the work while in Jorge
Dubcovsky's lab and paid his salary. JD was
funded by HHMI. DLC is funded by USDA-ARS
CRIS Project 2030-21430-014D. The funders had
no role in study design, data collection and
analysis, decision to publish, or preparation of the
manuscript.

**Competing interests:** The authors have declared
that no competing interests exist.

## Author summary

Daylength is an important environmental cue that plants and animals use to coordinate
important life history events with a proper season. In plants, timing of flowering to a par-
ticular season is an essential adaptation to many ecological niches. Perceiving changes in
daylength starts with the perception of light via specific photoreceptors such as phyto-
chromes. In temperate grasses, how daylength perception is integrated into downstream
pathways to trigger flowering is not fully understood. However, some of the components
involved in the translation of daylength perception into the induction of flowering in tem-
perate grasses have been identified from studies of natural variation. For example, specific
alleles of two genes called *EARLY FLOWERING 3* (*ELF3*) and *PHOTOPERIOD1* (*PPD1*)
have been selected during breeding of different wheat and barley varieties to modulate the
photoperiodic response to maximize reproduction in different environments. Here, we
show in the temperate grass model *Brachypodium distachyon* that the translation of the
light signal perceived by phytochromes into a flowering response is mediated by *ELF3*,
and that *PPD1* is genetically downstream of *ELF3* in the photoperiodic flowering pathway.
These results provide a genetic framework for understanding the photoperiodic response
in temperate grasses that include agronomically important crops such as wheat, oats, bar-
ley, and rye.

## Introduction

The transition from vegetative growth to flowering is an important developmental decision for
which the timing is often directly influenced by the environment (e.g. [1–4]). This critical life
history trait has been shaped over evolutionary time to enable reproduction to coincide with
the time of year that is most favorable for flower and seed development. Moreover, breeding to
adjust the timing of flowering in crops has been critical for adapting various crop varieties to
changing environments and to increase yield (e.g. [5]).

   In many plant species, changes in day-length and/or temperature provide seasonal cues
that result in flowering during a specific time of year [1,6]. Many temperate grasses such as
*Brachypodium distachyon* (*B. distachyon*), wheat, and barley that flower in the spring or early
summer months in response to increasing day-lengths are referred to as long-day (LD) plants
[7]. *B. distachyon* is closely related to the core pooid clade comprising wheat, oats, barley, and
rye and has a number of attributes that make it an attractive grass model organism suitable for
developmental genetics research [8,9].

   Variation in the LD promotion of flowering in temperate grasses such as wheat and barley
can be due to allelic variation at *PHOTOPERIOD1* (*PPD1*), a member of the pseudo-response
regulator (PRR) gene family (*PPD1* is also known as *PSEUDO RESPONSE REGULATOR 37*;
*PRR37*) [10,11]. Natural variation in *PPD1* resulting in either hypomorphic alleles as found in
barley or dominant *PPD1* alleles as found in tetraploid or hexaploid wheat impacts flowering
[10–15]. Specifically, natural recessive mutations in the conserved CONSTANS, CON-
STANS-LIKE and TIMING OF CAB EXPRESSION 1 (CCT) putative DNA binding domain
in the barley PPD1 protein cause photoperiod insensitivity and delayed flowering under LD
[11, 12], whereas wheat photoperiod insensitivity is linked to overlapping large deletions in the
promoter region of *PPD1* in either the A [13] or D genome homeologs [10]. These deletions
result in elevated expression of *PPD1*, particularly during dawn, causing rapid flowering even
under non-inductive SD conditions [13]. It is worth noting that although these wheat lines are
referred to as photoperiod insensitive (PI) varieties they still flower earlier under LD than

under SD if the timing of flowering is measured as the emergence of the wheat spike (heading time) [16]. It has been hypothesized that the large deletion within the *PPD1* promoter might remove a binding site for one or more transcriptional repressors [13]. To date, natural variation studies of flowering in *B. distachyon* have not pointed to allelic variation at *PPD1* and thus its role in LD flowering in *B. distachyon* is not known [17–21].

Variation in *EARLY FLOWERING 3* (*ELF3*; also known as *mat* and *eam*) impacts photoperiodic flowering in grasses, including wheat [22,23], barley [24,25], and rice [26]. In these plants, natural variation in *ELF3* allows growth at latitudes that otherwise would not be inductive for flowering, enabling these crops to be grown in regions with short growing seasons [5]. For example, *early maturity* (*eam*) loci have been used by breeders to allow barley to grow at higher latitudes in regions of northern Europe with short growing seasons [24,27]. The *eam8* mutant in the barley ortholog of *ELF3*, is a loss-of-function mutation that accelerates flowering under SD or LDs [24,25] similar to *elf3* loss-of-function alleles described previously in the eudicot model *Arabidopsis thaliana* (*A. thaliana*) [28]. Moreover, loss of function of *ELF3* in *B. distachyon* also results in rapid flowering under SD and LD, and expression of the *B. distachyon* ELF3 protein is able to rescue the *A. thaliana elf3* mutant, demonstrating a conserved role of *ELF3* in flowering across angiosperm diversification [29–31].

Work in *A. thaliana* has shown that *ELF3* is an important component of the circadian clock that acts as a bridge protein within a trimeric protein complex that also contains *LUX ARRTHYHMO* (LUX), and *EARLY FLOWERING 4* (*ELF4*) and is referred to as the evening complex (EC) [32]. Loss-of-function mutations in any of the proteins that make up the EC results in disrupted clock function and rapid flowering [33–36]. The peak expression of the EC at dusk is involved in the direct transcriptional repression of genes that make up the morning loop of the circadian clock including *A. thaliana PRR7* and *PRR9*, which are paralogs of grass *PPD1* and *PRR73* [37–39]. Recently, it has been shown that the EC also directly represses *PRR37*, *PRR95*, and *PRR73* in rice (*PRR37* is the rice ortholog of *PPD1*), indicating conservation of the role of the EC across flowering plant diversification [40]. Furthermore, *elf3* mutants in barley, wheat, and *B. distachyon* have elevated *PPD1* expression [23,24,30] indicating *ELF3* may impact flowering in part via *PPD1*, but to what extent remains to be determined.

The photoperiod and circadian pathways converge in the transcriptional activation of florigen/*FLOWERING LOCUS T1* (*FT1*) in leaves [6,41]. In temperate grasses, *PPD1* is required for the LD induction of *FT1*, whereas in *A. thaliana CONSTANS* (*CO*) is the main photoperiodic gene required for *FT1* activation in LD [16,42,43]. There are two *CO*-like genes in temperate grasses. Interestingly, in the presence of functional *PPD1*, *co1co2* wheat plants have a modest earlier heading phenotype suggesting they are in fact mild floral repressors, but in the absence of *PPD1*, *CO1* acts as a flowering promoter under LD [16]. To date, no null *co1co2* double mutants have been reported in *B. distachyon*. However, RNAi knock-down of *co1* results in a 30-day delay in flowering under 16h LD [44] and overexpression of *CO1* leads to earlier flowering in SD [44]. These results indicate that in *B. distachyon CO1* has a promoting role in flowering even in the presence of a functional *PPD1* gene, and suggest potential differences in the role of *CO1* in the regulation of flowering between *B. distachyon* and wheat.

Once FT1 is activated by LD it interacts with the bZIP transcription factor FD which triggers the expression of the MADS-box transcription factor *VERNALIZATION1* (*VRN1*) [45,46]. VRN1 in turn upregulates the expression of *FT1* forming a positive feedback loop that overcomes the repression from the zinc finger and CCT domain-containing transcription factor *VERNALIZATION2* (*VRN2*) [17,47–50]. The FT1 protein is then thought to migrate from the leaves to the shoot apical meristem, as shown in *A. thaliana* and rice [51,52], to induce the expression of floral homeotic genes including *VRN1*, thus converting the vegetative meristem to a floral meristem under favorable LD photoperiods.

Light signals are perceived initially by photoreceptors that initiate a signal transduction cascade impacting a variety of developmental responses to light [53]. The sensing of light is accomplished by complementary photoreceptors: phytochromes that perceive the ratio of red and far-red light, cryptochromes, phototropins, and Zeitlupe family proteins that detect blue light, and UV RESISTANCE LOCUS 8 that detects ultraviolet B light [54,55]. The phytochromes form homodimers that upon exposure of plants to red light undergo a confirmation shift to an active form causing the activation of a suite of downstream genes [54]. Exposure of plants to far-red or dark conditions causes reversion of the phytochromes to an inactive state [54].

There are three phytochromes in temperate grasses referred to as PHYTOCHROME A (PHYA), PHYTOCHROME B (PHYB), and PHYTOCHROME C (PHYC) [56]. Functional analyses of these phytochromes in temperate grasses revealed that PHYB and PHYC play a major role in the LD induction of flowering because loss-of-function mutations in either of these genes results in extremely delayed flowering [57–59] whereas loss-of-function mutations in *PHYA* in *B. distachyon* results in only a modest delay of flowering under inductive LD, indicating PHYB and PHYC are the main light receptors required for photoperiodic flowering in temperate grasses [29].The important role of PHYC in photoperiodic flowering is not universal as loss of *phyC* function in *A. thaliana* and rice only has small effects on flowering [60,61].

In temperate grasses, PHYB and PHYC are required for the transcriptional activation of a suite of genes involved in the photoperiod pathway, including *PPD1*, *CO1*, and *FT1*, and ectopic expression of *FT1* in the *B. distachyon phyC* background results in rapid flowering—a reversal of the *phyC* single-mutant, delayed-flowering phenotype [57,58,62]. Moreover, consistent with PHYB/C acting at the beginning of the photoperiodic flowering signal cascade, expression of genes encoding components of the circadian clock are also severely dampened in the *phyB* and *phyC* mutant backgrounds [29,57–59]. An exception to this is that the expression of *ELF3* is not altered in the temperate grass phytochrome mutants [29,58,59]. Recently, in *B. distachyon* it has been shown that PHYC can interact with ELF3, and this interaction destabilizes the ELF3 protein indicating that the regulation of ELF3 by PHY is at least in part at the protein level consistent with previous studies from *A. thaliana*, rice, and the companion study in wheat [23,29,40,63–66]. At present it is not clear to what extent the regulation of ELF3 by PHYs is critical for photoperiodic flowering.

Here, we show in *B. distachyon* by analyzing *phyC/efl3* double mutant plants that indeed the light signal perceived by phytochromes is mediated through ELF3 for photoperiodic flowering. The extreme delayed flowering of the *phyC* mutant disappears in the *phyC/elf3* double mutant which flower as rapidly as the *elf3* single mutant. Moreover, the expression profiles of genes in the photoperiod pathway are similar between *elf3* and *phyC/elf3* mutants compared to *phyC* mutants. Thus, *elf3* is completely epistatic to *phyC* at the phenotypic and molecular levels. Furthermore, we show strong, environment-dependent genetic interactions between *ELF3* and *PPD1*, which indicates that *PPD1* is a main target of ELF3-mediated repression of flowering. These results provide a genetic and molecular framework to understand photoperiodic flowering in the temperate grasses.

## Results

### Rapid flowering of *elf3* is epistatic to the delayed flowering of *phyC*

Previous studies in *B. distachyon* showed that PHYC can affect the stability of the ELF3 protein, and that the transcriptome of a *phyC* mutant resembles that of a plant with elevated ELF3 signaling [29]. Thus, it has been suggested that the extreme delayed flowering phenotype of the *phyC* mutant [58] could be mediated by ELF3 [29]. To test the extent to which the translation

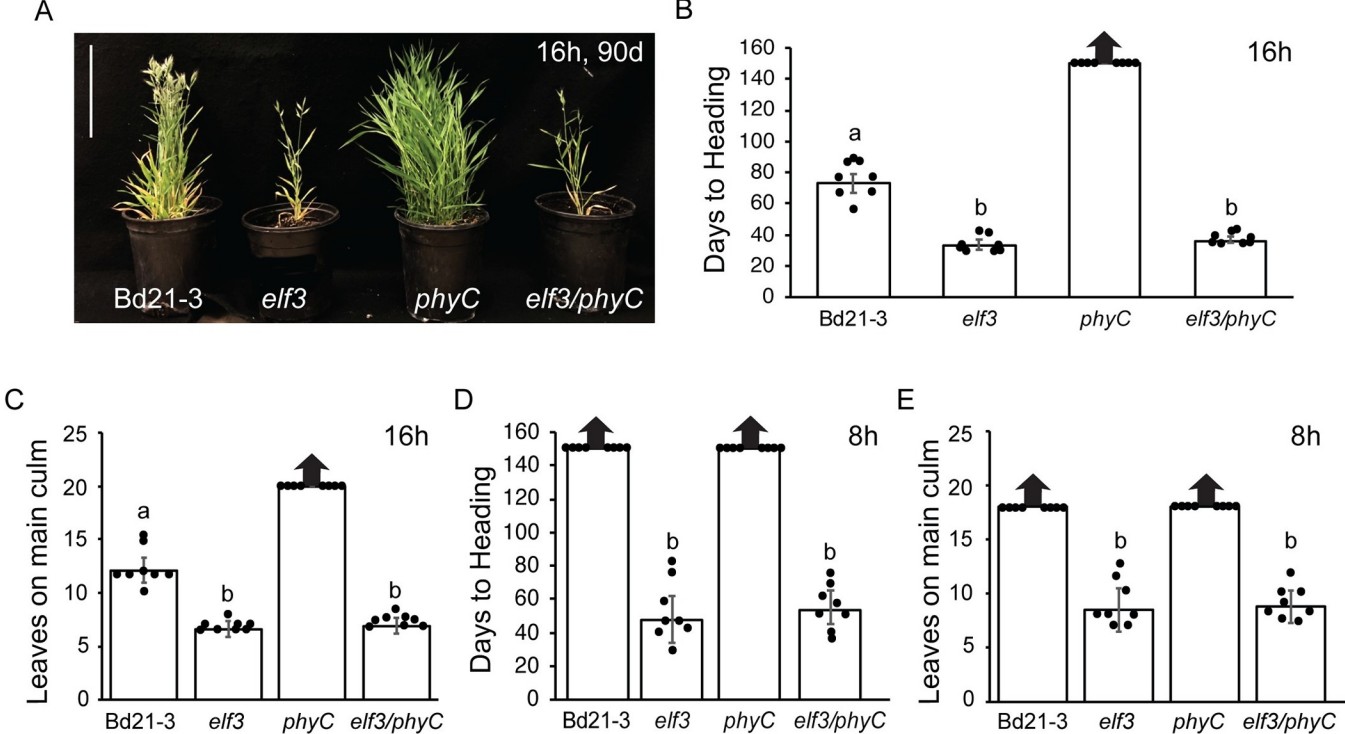

**Fig 1.** The rapid flowering of the *elf3* mutant is epistatic to the delayed flowering of the *phyC* mutant (**A**) Representative images of Bd21-3 wild-type, *elf3*, *phyC* and *elf3/phyC* double mutant plants grown in a 16h photoperiod at 90d after germination. Bar = 17cm. (**B, D**) Flowering times under 16h (**B**) or 8h daylengths (**D**) measured as days to heading of Bd21-3, *elf3*, *phyC*, and *elf3/phyC*. (**C**) Flowering phenotypes under 16h (**C**) or 8h daylengths (**E**) measured as the number of leaves on the parent culm at time of heading. Bars represent the average of 8 plants ± SD. Arrows above bars indicate that none of the plants flowered at the end of the experiment (150d). Letters (a, b) indicate statistical differences ($p < 0.05$) according to a Tukey's HSD test used to perform multiple comparisons.

of the light signal perceived by PHYC to control flowering is mediated by ELF3, we generated *elf3/phyC* double mutant plants and evaluated the flowering of the double mutant relative to that of *elf3* and *phyC* single mutants as well as Bd21-3 wild type under 16h-LD and 8h-SD (Fig 1).

Under both 16h LD and 8 SD photoperiods, we found that *elf3* is epistatic to *phyC*. Specifically, in LD *elf3/phyC* double mutants flowered rapidly by 38 days with 6.9 leaves similar to *elf3* mutants that flowered by 34 days with 6.6 leaves (Fig 1A and 1B). In contrast, *phyC* mutants had not flowered after 150 days with greater than 20 leaves when the experiment was terminated, and Bd21-3 wild-type flowered by 72 days with 12 leaves consistent with previous studies [49,58]. In 8h SD, *elf3/phyC* double mutants also flowered rapidly by 54 days with 8.8 leaves similar to *elf3* mutants that flowered by 48 days with 8.5 leaves (Fig 1D and 1E). In contrast, both Bd21-3 wild-type and *phyC* mutants had not flowered by 150 days with >18 leaves when the experiment was terminated (Fig 1D and 1E). These results indicate that the extreme delayed flowering mutant phenotype of *phyC* in *B. distachyon* is mediated by ELF3.

To determine if *PHYC* affects the expression of *ELF3*, we analyzed *ELF3* mRNA levels across a diurnal light cycle (16h light and 8h dark). There were no significant differences in *ELF3* expression at any time point in the *phyC* mutant relative to wildtype (S1 Fig) indicating that *PHYC* does not affect the transcriptional profile of *ELF3* in *B. distachyon* consistent with results from *A. thaliana* and wheat (67, 69).

## Effect of mutations in *PHYC* and *ELF3* on the transcriptional profiles of flowering time genes

To further understand how *PHYC* and *ELF3* affect flowering at a molecular level, we compared the mRNA levels of *B. distachyon* orthologs of the photoperiod and vernalization pathway genes *FT1*, *VRN1*, *PPD1*, *VRN2*, *CO1*, and *CO2* across a diurnal cycle in 16h LD in the *phyC* and *elf3* single mutants versus the *elf3/phyC* double mutant (Fig 2). We were particularly interested in determining how the expression profiles of "flowering-time genes" in the *elf3/phyC* double mutant compared to the *elf3* and *phyC* single mutant. The newly expanded fourth leaf was harvested for gene expression analyses because at this developmental stage in 16h daylengths the meristems of all of the plant genotypes are at a vegetative stage and thus are developmentally equivalent. Consistent with the rapid flowering of the *elf3* and *elf3/phyC* mutants, the mRNA expression levels of *FT1* and *VRN1* in these lines are significantly higher than the levels in wild-type and *phyC* mutants across all the time points tested (Fig 2A and 2D). Moreover, the overall expression profiles of *FT1* and *VRN1* in the *elf3* and *elf3/phyC* mutants were similar throughout the day. This is in contrast to the *phyC* mutant in which *FT1* and *VRN1* mRNA levels were lower than wild type throughout the day consistent with the delayed flowering phenotype of *phyC*. Although the elevated levels of *FT1* and *VRN1* in both *elf3* and *elf3/phyC* are consistent with their rapid flowering, the expression of the floral repressor, *VRN2*, exhibits a similar elevated expression profile throughout the day in both *elf3* and *elf3/phyC* relative to wild-type or *phyC* single-mutants (Fig 2E). The elevated *VRN2* expression levels in *elf3* mutant plants are consistent with previous results in *B. distachyon* and other grasses [23,29,30,40,64]. The transcriptional profile of *CO1* was similar in both the *elf3* and *elf3/phyC* mutants with elevated expression compared to wild-type between zt4-8 and then lower than wild-type between zt12-20 (Fig 2C). A similar expression pattern was found for *Hd1* (the rice

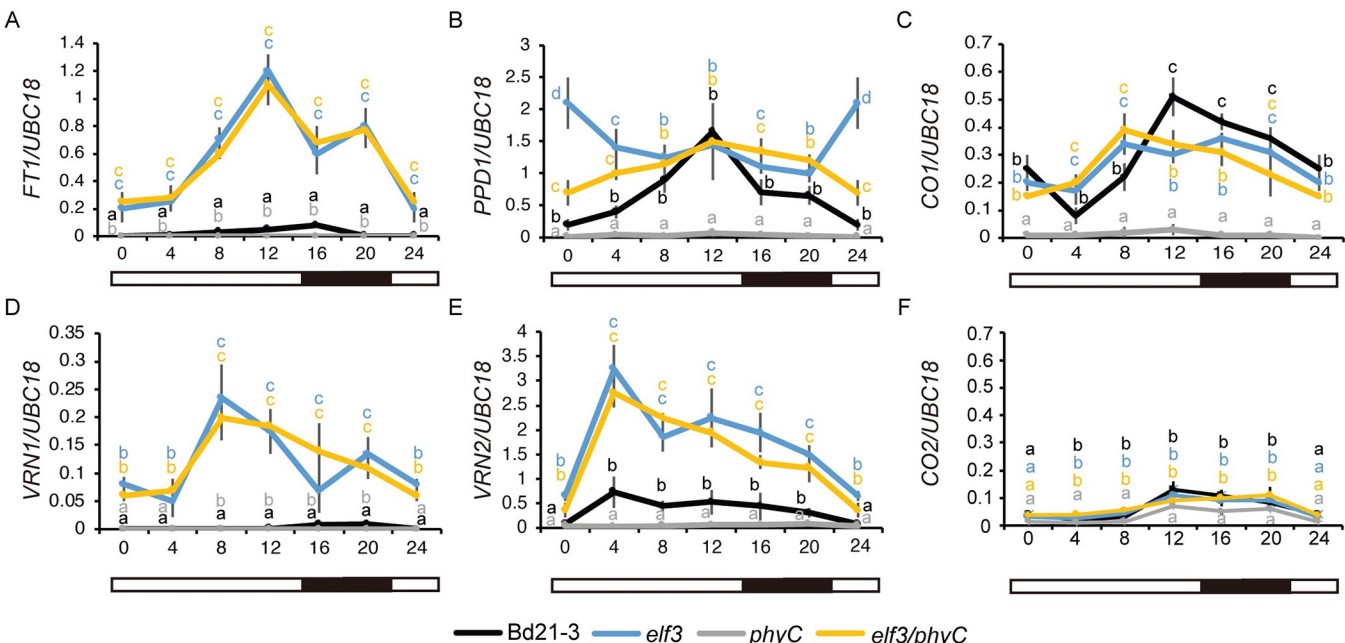

**Fig 2. Effect of loss-of-function mutations in *ELF3* and *PHYC* on the transcriptional profiles of six flowering time genes in 16h LD.** Normalized expression of (**A**) *FT1*, (**B**) *PPD1*, (**C**) *CO1*, (**D**) *VRN1*, (**E**) *VRN2*, and (**F**) *CO2* during a 24h diurnal cycle in Bd21-3 (black line), *elf3* (blue line), *phyC* (gray line) and *elf3/phyC* double mutant (orange line). Plants were grown in LDs until the fourth-leaf stage was reached (Zadoks = 14) at which point the newly expanded fourth leaf was harvested at zt0, zt4, zt8, zt12, zt16, and zt20. Note the zt0 value and zt24 value are the same. The average of four biological replicates is shown (three leaves per replicate). Error bars represent standard deviation of the mean. Data were normalized using *UBC18* as done in [49].

*CO* homolog) in the *elf3-1/elf3-2* double mutant in rice [40]. Consistent with previous reports, *CO1* expression levels remained low in *Brachypodium phyC* mutants throughout a diurnal cycle [58]. By contrast *CO1* expression is increased in the *phyC* mutants in wheat [57] indicating another difference in the regulation of *CO1* between these two species. Lastly, the *CO2* expression profiles were similar between wild-type, *elf3*, and *elf3/phyC*, whereas *CO2* mRNA levels were lower in *phyC* throughout a diurnal cycle (Fig 2F). In summary, the transcriptional profiles of *FT1*, *VRN1*, *VRN2*, *CO1*, and *CO2* are similar between *elf3* and *elf3/phyC* mutants consistent with *ELF3* acting downstream from *PHYC* in the photoperiod flowering pathway.

The transcriptional profile of *PPD1* indicates a more complex interaction between *PHYC* and *ELF3*. In wild type, the expression levels of *PPD1* peak at zt12 with the lowest expression level at dawn and during the evening consistent with previous reports of *PPD1* expression patterns in *B. distachyon* [29,30] (Fig 2B). In both the *elf3* and *elf3/phyC* mutants, we observed increased *PPD1* expression relative to wild-type at dawn and during the evening with expression levels similar to wild-type at zt12. Interestingly, the increased expression of *PPD1* observed at dawn and during the night in the *elf3/phyC* background was significantly lower than that of the *elf3* single mutant suggesting *PHYC* may impact *PPD1* expression via additional genes beyond *ELF3*. In contrast, *PPD1* expression levels were reduced in the *phyC* mutant relative to wild type, *elf3*, and *elf3/phyC* mutants throughout a diurnal cycle, consistent with the reduced *FT1* expression and delayed flowering phenotype of the *phyC* mutant.

## Identification and mapping of a *ppd1* mutant in *B. distachyon*

To determine the role of *PPD1* in flowering in *B. distachyon*, the genome-sequenced, sodium-azide mutant line NaN610 with a predicted high-effect mutation impacting a splice acceptor donor site in *PPD1* (BdiBd21-3.1G0218200) was obtained from the Joint Genome Institute (JGI) ([65]; **https://phytozome-next.jgi.doe.gov/jbrowse/**). A quarter of the NaN610 M3 seeds received were segregating for an extremely delayed flowering phenotype (Fig 3B–3D).

Due to the high mutant load of these NaN mutant lines, we validated through mapping that the delayed flowering phenotype is associated with *PPD1* (Fig 3E and 3F). We backcrossed NaN610 with Bd21-3 and confirmed a quarter of the plants in the BC1F2 population (n = 380) were delayed flowering, demonstrating the recessive nature of the mutant. Three Derived Cleaved Amplified Polymorphic Sequences (dCAPs) markers closely linked with *PPD1* were developed based on the variant's information for the NAN610 line, with one of the dCAPs primers located within the *PPD1* locus itself (Fig 3E and S1 Table). This approach allowed us to map the causative lesion to within a 1Mb interval (13.1Mb-14.2Mb) on the top arm of chromosome 1, demonstrating the delayed flowering phenotype is tightly linked with *PPD1* (Fig 3E).

To confirm that the predicted splice site mutation does in fact impact the splicing of *PPD1*, we sequenced the mRNA products of the *ppd1* NaN610 mutant line and Bd21-3 (Fig 3A). We found that the splice site mutation resulted in the mis-splicing of the sixth intron, generating a reading frame shift resulting in a truncated protein lacking the conserved CCT domain (Fig 3A). The extremely delayed flowering of the *B. distachyon ppd1* mutant is consistent with the *ppd1* null mutants described in wheat, which take >120 days to head under inductive LD conditions [16,43], demonstrating *PPD1* is required for LD flowering broadly within temperate grasses.

## Genetic interactions between *ELF3* and *PPD1* under long and short days

We and others have shown that *PPD1/PRR37* expression is increased in an *elf3* mutant background in *B. distachyon*, rice, and wheat (Fig 2B; [29,30,40,66]). Moreover, a CHIPseq analysis

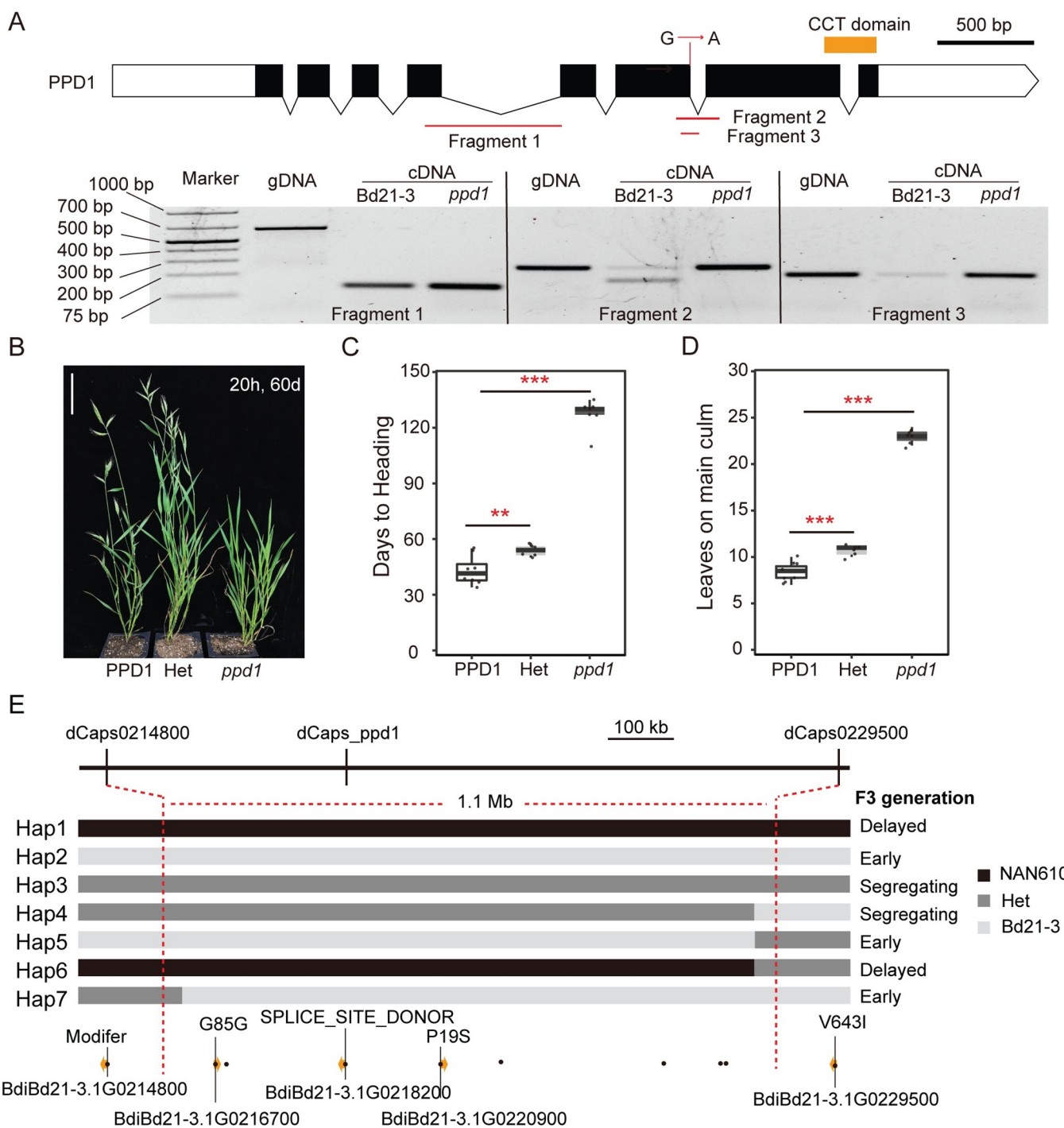

**Fig 3. Identification of a *ppd1* mutant.** (**A**) Gene structure of *PPD1* showing the location of the nucleotide change of the sodium azide-induced mutation; orange bar indicates the region that encodes the CCT domain. Below the gene structure diagram is a gel image of the reverse transcription polymerase chain reaction (PCR) (30 cycles of amplification) showing PCR products of *PPD1* cDNA in Bd21-3 and *ppd1* mutant plants. The location of primers used in each reaction are shown in the diagram above the gel image. (**B**) Representative photo of Bd21-3, heterozygous, and homozygous *ppd1* plants grown in a 20h LD. Picture was taken 60d after germination in 20h LD, bar = 5cm. (**C and D**) Flowering time was measured as days to heading (**C**) and the number of leaves on the parent culm at time of heading (**D**), ** indicates statistical differences (p < 0.01), *** indicates statistical differences (p < 0.001) by Student's t-test. (**E**) Fine mapping of *ppd1* in a population of 380 BC1F2 individuals. Individuals with seven different haplotypes were identified by three dCAPS markers and flowering times of each haplotype were determined in the F3 generation. Black, grey, and light grey rectangles represent NAN610, heterozygous, and Bd21-3 genotypes, respectively. Variants around the *PPD1* locus from the NAN610 line are shown with black dots, and yellow arrows indicate the coding genes within the mapped interval with the specific effect on the coding region indicated.

of ELF3 demonstrated that *PPD1* is directly bound by ELF3 in a time-of-day-responsive manner [29,40]. Thus, ELF3 acts as a direct transcriptional repressor of *PPD1* but the extent to which this explains the rapid flowering in the *elf3* mutant has not been tested. Therefore, we generated an *elf3/ppd1* double mutant to explore the genetic interactions of these two genes under a highly inductive 20h LD, inductive 16h LD, and non-inductive 8h SD (Fig 4).

Under all photoperiods, the *elf3/ppd1* double mutant flowered significantly later than the *elf3* single mutant (Fig 4). Interestingly, under 20h LD, the *elf3/ppd1* double mutant flowered earlier than *ppd1* by 16.2 days forming 3.0 fewer leaves whereas under 16h days *elf3/ppd1* mutant flowered significantly later than *ppd1* by 13.7 days with 2.5 more leaves. In 8h SD, only *elf3* mutant plants were able to flower; Bd21-3, *ppd1*, and *elf3/ppd1* all failed to flower by the end of the experiment. It is also worth noting that *elf3/ppd1* double mutants are still able to respond to different photoperiods, with longer days resulting in significantly earlier flowering plants than under shorter days (Fig 4B and 4E and 4H). These results indicate that there are strong genetic interactions between *ELF3* and *PPD1* under different photoperiods, that *PPD1* is a key flowering regulator downstream of *ELF3*, and that there is a residual photoperiodic response that is independent of these two genes.

## Effect of mutations in *ELF3* and *PPD1* on the transcriptional profiles of flowering time genes

To understand how *ELF3* and *PPD1* affect flowering at a molecular level, we measured the mRNA levels of *FT1, VRN1, PPD1, VRN2, CO1,* and *CO2* in the *elf3* and *ppd1* single mutants and the *elf3/ppd1* double mutant across a diurnal cycle in 16h LD (Fig 5). As noted before, *FT1* and *VRN1* expression levels were elevated in the *elf3* mutant background; however, in the *elf3/ppd1* double mutant, expression of these genes remained low and resembled the expression profile of *ppd1* single mutants (Fig 5A and 5D). The low expression levels of *FT1* and *VRN1* in *ppd1* and *ppd1/elf3* mutants is consistent with the delayed flowering phenotype of both of these mutants in 16h LD. The *VRN2* expression profile was similar between wild type and *ppd1* mutant plants with low expression levels at dawn and increased expression throughout the light cycle before expression levels dropped in the dark (Fig 5E). Interestingly, *VRN2* expression levels are similarly elevated throughout the day in *elf3* and *elf3/ppd1* mutants compared to wild type (Fig 5E).

Consistent with the expression patterns of *PPD1* in wild type and *elf3* shown in Fig 2, the expression levels of *PPD1* peak at zt12 in wild-type and the *elf3* mutant has increased *PPD1* expression relative to wild type at dawn and during the evening (Fig 5B). *PPD1* expression levels in the *ppd1* mutant should be interpreted with caution because we do not know the effect of the splice site mutation on the mRNA stability. Significantly higher levels of *PPD1* expression were observed in *ppd1* relative to wild type at ZT8 and ZT16, and in *elf3/ppd1* relative to *elf3* at dawn. However, the expression patterns of *PPD1* were most similar between *ppd1* and wild type and between *elf3/ppd1* and *elf3* (Fig 5B).

*CO1* and *CO2* expression both exhibit peak expression in wild type at zt12 with expression dampening in the evening consistent with previous reports [29,58]. Interestingly, expression levels of *CO1* and *CO2* were elevated between zt4-8 in *elf3* compared with wild-type. However, at zt16 and zt20, expression levels were similar in wild type, *elf3*, and *elf3/ppd1* mutants, whereas at zt12 the expression of *CO1* was reduced in *elf3* compared to wild type. In contrast, the expression levels of *CO1* and *CO2* were lowest in *ppd1* compared to the other lines at zt8. In the *elf3/ppd1* mutants, *CO1* and *CO2* expression was most similar to *ppd1* in the morning and most similar to *elf3* in the evening (Fig 5C and 5F). These results indicate complex interactions between PPD1 and ELF3 in the regulation of *CO1* and *CO2*.

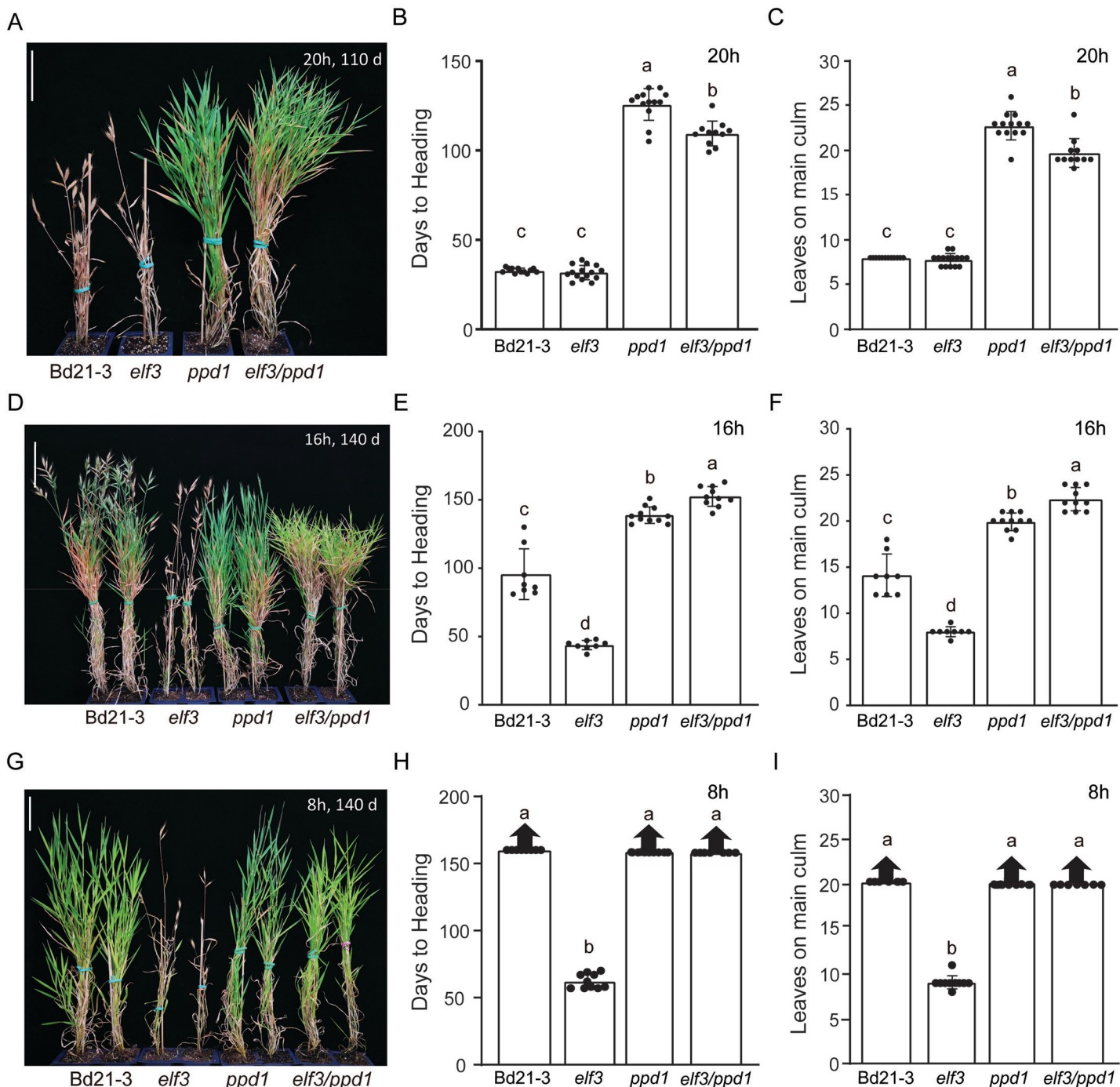

**Fig 4. Genetic interactions between the delayed flowering *ppd1* mutant and the rapid flowering *elf3* mutant.** Representative image of Bd21-3 wild-type, rapid flowering *elf3* mutant, delayed flowering *ppd1* mutant, and delayed flowering *elf3/ppd1* double mutant grown in a 20h photoperiod (**A**), 16h photoperiod (**D**), and 8h photoperiod (**G**). Picture was taken after 110d, for the 20h LD (**A**) and 140d after germination for the 16h LD and (**D**) 8h SD. Scale bar = 5cm. (**B**, **E**, **H**) Flowering times under 20h (**B**), 16h (**E**), 8h (**G**) measured as days to heading of Bd21-3, *elf3*, *ppd1*, and *elf3/ppd1*. Flowering times under 20h (**C**), 16h (**F**), and 8h (**I**) measured as the number of leaves on the parent culm at time of heading. The 8h experiment was repeated three times. The first experiment resulted in *ppd1* plants that stopped producing new leaves before wild type. One possibility for the cessation of new leaf production in *ppd1* plants in this experiment is that the meristem transitioned to flowering, but then did not proceed to heading. However, in two subsequent experiments *ppd1* plants continually produced new leaves for the duration of the experiment similar to wild type and this data is shown in (**I**). Data for all three experiments are shown in S1 Data for Fig 4. When grown under non-inductive conditions for 120 days or more, a few *B. distachyon* plants flower; we consider this a stochastic flowering response because the majority of plants do not flower. Bars represent the average of 8 plants ± SD. Arrows above bars indicate that none of the plants flowered at the end of the experiment (150d, >20 leaves). Letters (a, b, c, d) indicate statistical differences (p < 0.05) according to a Tukey's HSD test used to perform multiple comparisons.

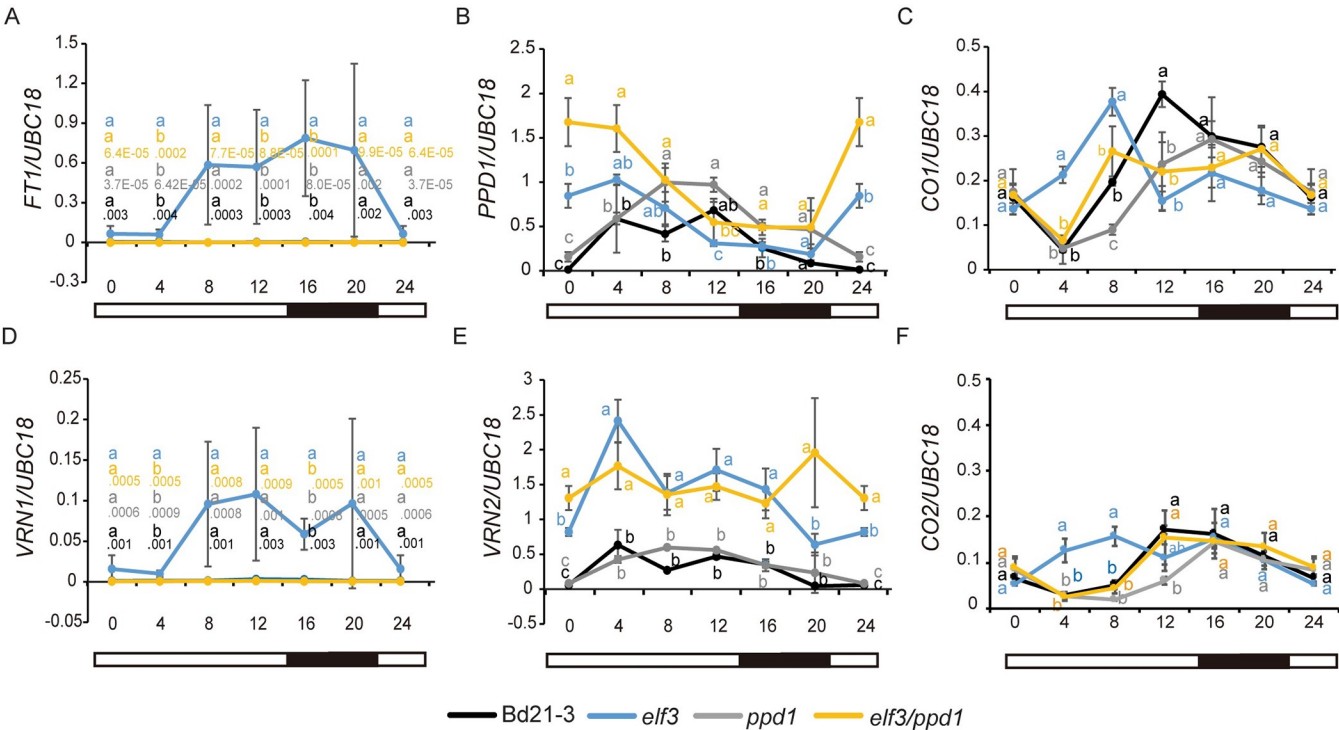

**Fig 5. Effect of loss-of-function mutations in *ELF3* and *PPD1* on the transcriptional profiles of six flowering-time genes in 16h LD.** The fourth newly expanded leaves were harvested every 4h over a 24-hour period; three biological replicates (two leaves per replicate) were harvested at each time point for each genotype. Diurnal expression of *FT1* (**A**), *PPD1* (**B**), *CO1* (**C**), *VRN1* (**D**), *VRN2* (**E**), and *CO2* (**F**) were detected in Bd21-3 (black line), *elf3* (blue line), *ppd1* (grey line) and *elf3/ppd1* double (orange line). Bars represent the average of three biological replicates ± SD. Letters (a, b, c, d) indicate statistical differences (p < 0.05) according to a Tukey's HSD test used to perform multiple comparisons, letter color corresponds to the four different genotypes. The black, gray, and orange lines overlap given the scale used to show *ELF3* expression in the same graph; the orange line is arbitrarily shown on top. Specific expression values are shown below the lettered statistical test. Raw data is included in S1 Data file.

## Constitutive expression of *ELF3* results in delayed flowering and lower *PPD1*, *FT1*, and *VRN1* expression levels

In our previous study, we showed that overexpression of *ELF3* in the *elf3* mutant background results in strongly delayed flowering ([30], Fig 6A). However, this was done in the T0 generation, so we evaluated the flowering time and expression of downstream flowering-time genes in the T1 generation. We grew four *UBI::ELF3/elf3* transgenic lines alongside Bd21-3 and *elf3* in a 16h photoperiod, and harvested the newly expanded fourth leaf at zt4. This time point was chosen because expression of several critical genes such as *CCA1*, *TOC1*, *LUX*, *PPD1*, *VRN2*, *CO1*, and *CO2* were significantly different in the morning in the *elf3* single mutant compared with wild-type [29,30]; Figs 2 and 5). We first confirmed that all of the *UBI::ELF3/elf3* transgenic lines had elevated *ELF3* mRNA levels, and found indeed there is a significant increase of *ELF3* expression in the transgenic lines (Fig 6C). To understand how *UBI::ELF3/elf3* affects flowering, we evaluated expression levels of *FT1*, *VRN1*, *PPD1*, *VRN2*, *CO1*, and *CO2* in wild type, *elf3*, and *UBI::ELF3/elf3*. Consistent with the delayed flowering, *FT1* and *VRN1* expression levels of *UBI::ELF3/elf3* were reduced relative to wildtype compared to elevated levels *elf3* relative to wildtype (Fig 6D and 6G). Also, the expression of *PPD1*, *VRN2*, *CO1*, and *CO2* were decreased in *UBI::ELF3/elf3*, indicating ELF3 is playing a broad repressive role in regulating CCT domain containing genes responding to photoperiodic flowering.

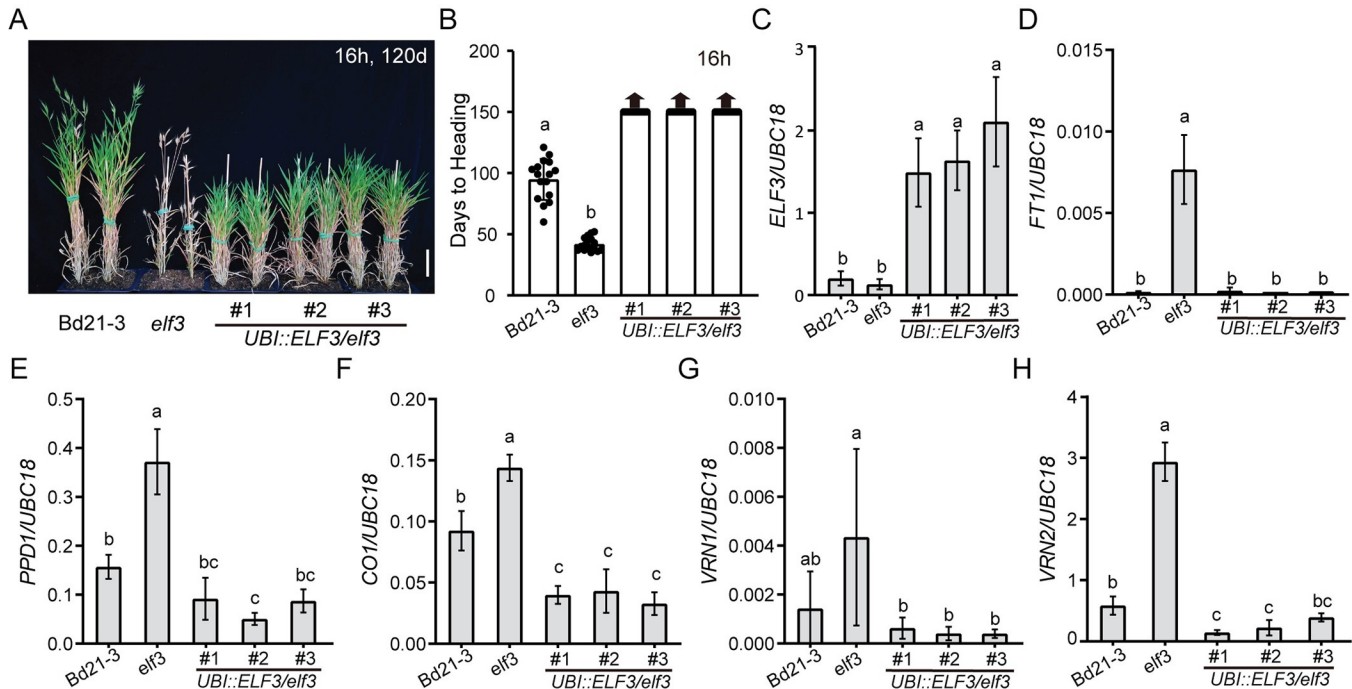

**Fig 6. Overexpression of *ELF3* in the *elf3* mutant delays flowering.** (**A**) Representative image of Bd21-3 wild type, *elf3*, and three independent transgenic lines of *UBI::ELF3* in the *elf3* background grown in a 16h photoperiod. Images were taken 120d after germination. Bar = 5 cm. The fourth newly expanded leaves were harvested at zt4 in 16h. (**B-I**), Normalized expression of *ELF3* (**C**), *FT1* (**D**),*PPD1* (**E**), *CO1* (**F**), *VRN1* (**G**), *VRN2* (**H**) in Bd21-3 wild type, *elf3*, and three *UBI::ELF3/elf3* transgenic lines. Expression of *CO2* is shown in S2 Fig. Bars represent the average of four biological replicates ± SD.

## Discussion

### The phytochromes *PHYC/PHYB* and *ELF3* connection

*B. distachyon* has an obligate requirement for LD to flower [8,49,67]. Previous studies have shown the important roles that both PHYC and ELF3 play in photoperiodic flowering in *B. distachyon* [29,30,58]. Specifically, mutations in *phyC* result in extremely delayed flowering whereas loss-of-function mutations in *elf3* result in rapid flowering in either LD or SD [30,58]. Furthermore, *phyC* mutants resemble plants grown in SD both morphologically and at the transcriptomic level regardless of day-length whereas *elf3* mutants resemble plants grown in LD both morphologically and at the transcriptomic level regardless of day-length [29,30,58]. Thus, we were interested in exploring the genetic relationships between *PHYC* and *ELF3*. The extreme delayed flowering phenotype observed in *phyC* mutant plants is mediated by *ELF3* because *phyC/elf3* double mutants flower rapidly in LD and SD similar to *elf3* mutants. Similar genetic interactions between *phyB* and *elf3* were also found in wheat in the companion study [66], suggesting these interactions are likely to be conserved broadly in temperate grasses. Loss-of-function mutations in *phyB* in wheat also result in delayed flowering similar to *phyC* [59]. At present, no null *phyB* alleles have been reported in *B. distachyon;* however, PHYB is able to heterodimerize with PHYC in *B. distachyon* and wheat [29, 57], and both *phyB* [59] and *phyC* [57] mutants are extremely late flowering in wheat suggesting that both PHYs are likely required for photoperiodic flowering in the temperate grasses, perhaps because PHYB/PHYC heterodimers are required for flowering regulation.

Phytochrome regulation of ELF3 at the post-translational level rather than at the transcriptional level is likely to be the critical interaction impacting flowering. In *A. thaliana*, *B.*

*distachyon*, and wheat, *phyB/phyC* mutants do not impact the circadian oscillation of *ELF3* mRNA levels [29,59,68]. However, in all three species PHYB and PHYC have been shown to interact with the ELF3 protein, but the stability of the ELF3 protein upon exposure to light differs between *A. thaliana* and temperate grasses [29,63,66,69]. Specifically, in *A. thaliana*, PHYB contributes to the stability of the ELF3 protein during light exposure leading to ELF3 accumulation at the end of the day [63,68], whereas in rice ELF3 is degraded and or modified during light exposure in a PHY-mediated process [40]. In temperate grasses, ELF3 protein accumulates during the night and is rapidly degraded or modified upon light exposure [29,67], and this is likely to be a PHY mediated response as well.

The differences in how phytochromes impact the stability of the ELF3 protein might explain the contrasting flowering phenotypes of the *phyB/phyC* mutants between *A. thaliana* and temperate grasses. In *A. thaliana*, *phyB* mutants flower more rapidly than wild type in either LD or SD and *phyC* mutants flower earlier under SD [60], whereas in temperate grasses *phyB* or *phyC* mutants are extremely delayed in flowering [57–59]. However, ELF3 acts as a flowering repressor in both *A. thaliana* and grasses [30,32]. In *A. thaliana* PHYB stabilizes the ELF3 protein; therefore, in *phyB* mutants, ELF3 is no longer stable leading to rapid flowering. In contrast, in temperate grasses and rice, in the absence of *phyB* or *phyC* the ELF3 protein is more stable leading to delayed flowering.

Interestingly, overexpression of ELF3 results in extremely delayed flowering in *B. distachyon* [29,30] (Fig 6A and 6B). Given that the regulation of ELF3 appears to occur at the protein level, one might not expect that overexpression would cause such a strong flowering delay. However, if the ELF3 protein is expressed at a high level such that the degradation machinery is unable to degrade much of the ELF3 protein during LD, then a strong flowering delay might occur. In support of this idea, the delayed flowering of overexpression of ELF3 is mitigated when plants are grown under constant light versus 16h LD [29]. It is worth noting that although overexpression of ELF3 generally leads to delayed flowering in different plant species, there is considerable variation in the magnitude of this delayed flowering [29,63,66] (Fig 6A and 6B).

Similar genetic interactions between *ELF3* and *PHYB* have also been observed in rice which is a SD-flowering plant that has two rice-specific *ELF3* paralogs [70]. Mutations in either paralog results in delayed flowering in SD or LD in contrast to the rapid flowering observed in temperate grasses containing *elf3* mutations [40,71,72]. Also in contrast to the situation in temperate grasses, *phyB* mutants flower more rapidly than wild type in rice [73]. Despite the flowering differences of the *elf3* and *phyB* mutants between rice and temperate grasses, the flowering phenotype of *phy* mutants is ELF3 mediated because in both rice and temperate grasses *elf3* is epistatic to *phyB or phyC* [Fig 1; 40, 67]. Moreover, PHYB and ELF3 proteins interact impacting the modification of ELF3 by light [40]. The opposite roles that phytochromes and *elf3* have on flowering in rice and temperate grasses is likely due, at least in part, to the reverse role that the downstream *PPD1/PRR37* gene has on flowering. *PPD1* is a promoter of flowering in LD temperate grasses but is a repressor of flowering in SD grasses such as rice [11,16,74,75] (Fig 3).

## The *ELF3* and *PPD1* connection

The extremely delayed flowering of *B. distachyon ppd1* mutant plants under LD is similar to the extremely delayed heading of *ppd1* mutants in wheat [16]. However, a previous study in *B. distachyon* using a CRISPR induced *ppd1* mutant allele which has a 1bp deletion in the sixth exon of *PPD1* has a milder delayed flowering phenotype with plants taking around 40 days to flower under 20h LD, whereas the mutant *ppd1* plants presented here flower around 120 days

in 20h LD [29]; Figs 3 and 4). In both studies, wild-type Bd21-3 plants flower on average between 25–30 days in 20h LD consistent with previous reports in *B. distachyon* [19,49,76]. The differences in flowering time between the two *B. distachyon ppd1* mutant alleles suggests that the CRISPR induced *ppd1* allele is a weaker hypomorphic allele than the *ppd1* mutant allele characterized in this study. This is further supported by the fact that the *ppd1* allele described here has an extremely delayed flowering phenotype similar to the null *ppd1* wheat allele [16].

The *ppd1/elf3* double mutant is delayed in flowering relative to *elf3* mutant plants indicating that *PPD1* is downstream of ELF3 in photoperiodic flowering. This is also consistent with the elevated *PPD1* expression levels observed at dawn and dusk in the *elf3* mutant relative to wild-type in temperate grasses [30,70] (Fig 5). Indeed, ELF3 binds to the *PPD1/PRR37* promoter in *B. distachyon*, wheat, and rice indicating ELF3 is a direct transcriptional repressor of *PPD1/PRR37* in grasses [29,40,66]. ELF3 does not have any known DNA binding activity and thus, the direct repression is likely to be due to ELF3's interaction with a LUX transcription factor which, from studies in *A. thaliana*, recognizes GATWCG motifs that are also found in the *PPD1* promoter in grasses [37,38,67]. Interestingly, photoperiod insensitivity in wheat is associated with deletions in the *PPD1* promoter, which remove the LUX binding site and results in elevated *PPD1* expression at dawn similar to the *PPD1* expression dynamics observed in *elf3* and *lux* mutant plants [10–13,39,77,78]. In the companion wheat paper, ChIP-PCR experiments show ELF3 enrichment of the DNA region around the LUX binding site in the *PPD1* promoter, which is present within the region deleted in photoperiod-insensitive wheats. These results demonstrate that removal of the evening complex binding site leads to elevated expression in *PPD1* and accelerated heading under SD in many photoperiod-insensitive wheats [66].

The characterization of *elf3/ppd1* mutant plants under different photoperiods reveal complex interactions between the two genes and their downstream targets depending on the environment. For example, *elf3/ppd1* mutant plants are earlier flowering than *ppd1* mutant plants under 20h day-lengths, but are later flowering under 16h and 8h day-lengths. This is in contrast to *elf3/ppd1* mutants in wheat, which head earlier than *ppd1* under a 16h day-lengths indicating that *ELF3* can delay heading independently of PPD1 in this condition [66]. Thus, there are differences between *B. distachyon* and wheat in the effects of *ELF3* on heading in the absence of *PPD1*. We speculate that these differences may be related to the different interactions observed between *CO1* and other flowering genes (e.g. PHY) in these two species. For example, in wheat *phyC* and *ppd1* mutants, *CO1* expression levels are elevated compared to wild type, whereas in *B. distachyon CO1* expression is reduced in both mutants [57,58,66].

## Materials and methods

### Plant materials and growth conditions

The rapid flowering mutant *elf3* and four *UBI::ELF3/elf3* transgenic lines in *B. distachyon* were previously characterized [30] as was the delayed flowering *phyC* mutant [58]. All the mutants used for phenotyping and expression in this study were backcrossed at least twice with the wild-type Bd21-3 accession. Seeds were imbibed in the dark at 5°C for three days before planting in soil. Three photoperiods 8h-SD (8h light/16h dark), 16h-LD (16h light/ 8h dark), and 20h-LD (20h light/ 4h dark) were used. For Figs 1 and 2, plants were grown at the University of California-Davis in growth chambers with metal halide and sodium bulbs as the light source and a temperature of 22°C during light periods and 17°C during dark periods. Light intensity was approximately 300umol m-2s-1 at plant height. For Figs 4 and 5, plants were grown at the University of Wisconsin-Madison in growth chambers with T5 fluorescent bulbs (5000 K), and light intensity was approximately 200umol m-2s-1 at plant height. Temperatures averaged

22˚C during light periods and 18˚C during dark periods. Flowering time was estimated by measuring days to heading and leaves on the main culm at time of heading. Recording of days to heading was done as the days from seed germination to the first emergence of the spike.

## Generation of *elf3/phyC* and *elf3/ppd1* double-mutant lines

Epistasis analysis between *phyC*, *ppd1*, and *elf3* was studied by generating *elf3/phyC* and *elf3/ppd1* double mutants. *phyC* mutants were crossed with *elf3* and *elf3/phyC* homozygous double mutant plants were selected in a segregating F2 population by genotyping using primers in S1 Table. Similarly, *ppd1* was crossed with *elf3*, and *elf3/ppd1* homozygous double mutant individuals were selected by genotyping using primers in S1 Table in the segregating F2 population. Flowering time of *elf3/ppd1* double mutant were estimated by growing with Bd21-3, *elf3*, *ppd1* side by side in 8h SD, 16h LD, and 20h LD, and *elf3/phyC* double mutant plants were grown in 8h SD and 16h LD.

## RNA extraction and qPCR

The method for RNA extraction, cDNA synthesis and quantitative PCR (qPCR) is described in [49]. Primers used for gene expression analyses are listed in S1 Table.

## Statistical analyses

Comparison of more than two genotypes were performed by using *agricolae* package in R [79]. Statistically significant differences among different genotypes were calculated by using one-way analysis of variance (ANOVA) followed by a Tukey's HSD test. Student's t-test was used for analyzing the difference between two genotypes, significant if $P < 0.05$.

## Supporting information

**S1 Fig. Effect of loss of function mutations in *PHYC* on the transcriptional profile of *ELF3*.**
(TIF)

**S2 Fig. Normalized expression of *CO2* detected in Bd21-3, *elf3*, and three *UBI::ELF3/elf3* transgenic lines.**
(TIF)

**S1 Table. Primers used in this study.**
(PDF)

**S1 Data. S1_Data.xlsx file contains all raw data used in this study.**
(XLSX)

## Acknowledgments

Thanks to Frédéric Bouché for fruitful discussions about photoperiod sensing in *B. distachyon*.

## Author Contributions

**Conceptualization:** Daniel P. Woods, Weiya Li, Jorge Dubcovsky, Richard M. Amasino.

**Data curation:** Daniel P. Woods, Weiya Li.

**Formal analysis:** Daniel P. Woods, Weiya Li, Jorge Dubcovsky, Richard M. Amasino.

**Funding acquisition:** Jorge Dubcovsky, Richard M. Amasino.

**Investigation:** Daniel P. Woods, Weiya Li.

**Project administration:** Richard M. Amasino.

**Resources:** Daniel P. Woods, Weiya Li, Richard Sibout, Mingqin Shao, Debbie Laudencia-Chingcuanco, John P. Vogel.

**Supervision:** Daniel P. Woods, Weiya Li, Jorge Dubcovsky, Richard M. Amasino.

**Validation:** Weiya Li.

**Visualization:** Daniel P. Woods, Weiya Li.

**Writing – original draft:** Daniel P. Woods.

**Writing – review & editing:** Daniel P. Woods, Weiya Li, Richard Sibout, Mingqin Shao, Debbie Laudencia-Chingcuanco, John P. Vogel, Jorge Dubcovsky, Richard M. Amasino.

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
