## [Decision Letter · Decision Letter 0]

10 Dec 2022

Dear Dr Woods,

Thank you very much for submitting your Research Article entitled 'PHYTOCHROME C regulation of PHOTOPERIOD1 is mediated by EARLY FLOWERING 3 in Brachypodium distachyon' to PLOS Genetics.

The manuscript was fully evaluated at the editorial level and by independent peer reviewers. The reviewers appreciated the attention to an important topic but identified some concerns that we ask you address in a revised manuscript.

We therefore ask you to modify the manuscript according to the review recommendations. Your revisions should address the specific points made by each reviewer.

Yours sincerely,

Claudia Köhler

Section Editor

PLOS Genetics

Claudia Köhler

Section Editor

PLOS Genetics

Reviewer's Responses to Questions

**Comments to the Authors:**

Reviewer #1: This manuscript examined the genetic relationships between PHYC, ELF3 and PPD in their role in determining flowering time in B. distachyon. Previous work in B. distachyon showed a PHYC-ELF3 interaction and a reduction in ELF3 protein abundance under light irradiation in B. distachyon. The work in this paper examines the flowering time phenotypes of elf3, phyc and elf3phyc mutants to establish that the rapid flowering of the elf3 mutant is epistatic to the delayed flowering of the phyC mutant.

Gene expression of select flowering time related genes are examined in these same backgrounds, with the conclusion that the transcriptional profiles of FT1, VRN1, VRN2, CO1, and CO2 are similar between elf3 and elf3/phyC mutants consistent with ELF3 acting downstream from PHYC in the photoperiod flowering pathway. A novel ppd mutant was identified and analyzed molecularly, and concluded to be a splice site mutation, with uncertain effects on truncated protein abundance, but likely a null based on comparison to a wheat ppd mutant.

Further experiments with this mutant followed the template used for phyC: flowering time phenotypes of ppd, elf3 and ppd elf3 double mutant descriptions. The ppd mutant is largely epistatic to elf3 with respect to flowering time, but there is more to the story that is unclear. Even more so with respect to the transcriptional profiles of FT1, VRN1, VRN2, CO1, and CO2, which varied with the gene of interest. ppd was epistatic to elf3 with respect to FT levels, but other results indicated complex interactions between PPD1 and ELF3 in the regulation of CO1 and CO2.

Overall the results are largely supportive of the conclusions the authors wish to draw, but lack any type of molecular analysis that would confirm the conclusions drawn from genetics alone. Are ELF3 protein levels dependent on PHYC levels? This should be tested in their phyC mutant. ELF3 transcript levels could also be affected by the phyC mutant, and transcript levels should be tested as well.

Reference to a companion paper was made, but was not available to this reviewer.

The authors reference a bioRxiv manuscript which, in many ways, goes beyond this paper, in providing evidence for a PHYC/ELF3 complex at gene promoters. However, the above suggestions of the potential of ELF3 levels (protein and/or transcript) regulated by PHYC would greatly strengthen this work.

Additional points:

Why is the peak time of CO2 in the elf3 mutant so different between Figure 2F and Figure 5F? This is the same time course with the same mutant background so results should not be so divergent. Something needs to be repeated or explained, as swapping the elf3 data between the two figures (each coming from 3 biological reps?) would lead to different conclusions. Also, error bars are massive for A and D in Figure 5 for elf3.

P. 20: “….elf3 mutant compared with wild-type however from zt 12-20 expression levels were lower than wild-type with a profile similar to elf3/ppd1 mutant plants.” Based on stats elf3 and WT are not different for CO2 between ZT 12-20, just CO1.

Figure 4H shows 3 lines not flowering (heading) at all but 4I indicates number of leaves at time of heading in two of those lines [ppd1 and elf3ppd1[]. How can that be? Clarify. Also in legend, must mean:

Flowering times under 20h (B), 16h (E), 8h (H) measured as days to heading of Bd21-3,

elf3, ppd1, and elf3/ppd1.

On page 21; Fig 6B should be Fig 6C. Fig 6C and F should be Fig 6D and G.

On page 24; Fig 6 should be Fig 6A and B.

On page 25; Fig 6 should be Fig 6A and B.

Reviewer #2: The authors provide a concise genetic analysis of photoperiodic flowering in the temperate grass Brachypodium. Through the phenotypic and molecular analysis of single and double mutants they show that ELF3 acts downstream of PHYC in the induction of the flowering activator PPD1, with ELF3 being a negative regulator of PPD1 expression. For this study, the authors identified a ppd1 mutant which was not available before.

The authors´conclusions are supported by the data. The manuscript is well-written. I only have a few minor comments:

Fig. 4H,I. Fig. 4H shows that the mutants ppd1 and elf3/ppd1 fail to head within the time frame of the experiment. Nevertheless, the authors show in Fig. 4H the number of leaves at the time of heading. That seems to be a discrepancy to me. Can you please explain?

Fig. 5A, D: The transcript levels of FT1 and VRN1 cannot be seen in the WT and ppd1 mutants I assume that they are not seen because they are as low as in the elf3/ppd1 mutant. Can you please indicate that or make them visible in another way? On the other hand, would it not be expected that FT1 and VRN1 transcript levels in the WT are as high as in the elf3 mutant in LD because these genotypes flower at the same time in LD (see Fig. 4a)?

**Have all data underlying the figures and results presented in the manuscript been provided?**

Reviewer #1: Yes

Reviewer #2: None

PLOS authors have the option to publish the peer review history of their article (what does this mean?). If published, this will include your full peer review and any attached files.

Reviewer #1: No

Reviewer #2: No

---

## [Editor Report · Decision Letter 1]

17 Mar 2023

Dear Dr Woods,

We are pleased to inform you that your manuscript entitled "PHYTOCHROME C regulation of photoperiodic flowering via PHOTOPERIOD1 is mediated by EARLY FLOWERING 3 in Brachypodium distachyon" has been editorially accepted for publication in PLOS Genetics. Congratulations!

Yours sincerely,

Claudia Köhler

Section Editor

PLOS Genetics

Comments from the reviewers (if applicable):

**Data Deposition**

http://datadryad.org/submit?journalID=pgenetics&manu=PGENETICS-D-22-01183R1

**Press Queries**

---

## [Editor Report · Acceptance letter]

25 Apr 2023

PGENETICS-D-22-01183R1 

PHYTOCHROME C regulation of photoperiodic flowering via * PHOTOPERIOD1* is mediated by *EARLY FLOWERING 3* in <i>Brachypodium distachyon<i> 

Dear Dr Woods, 

We are pleased to inform you that your manuscript entitled "PHYTOCHROME C regulation of photoperiodic flowering via * PHOTOPERIOD1* is mediated by *EARLY FLOWERING 3* in <i>Brachypodium distachyon<i>" has been formally accepted for publication in PLOS Genetics! Your manuscript is now with our production department and you will be notified of the publication date in due course.

With kind regards,

Zsofia Freund

PLOS Genetics

On behalf of:
